# Identifying Different Sources of the Benefit: Simulation of DRT Operation in the Heartland and Hinterland Regions

Hyunmyung Kim [1,2], Jaeheon Choi [1,*], Sungjin Cho [3,*], Feng Liu [4], Hyungmin Jin [1], Suhwan Lim [1], Dongjun Kim [1], Jun Lee [5] and Chang-Hyeon Joh [6]

1 Studio Galilei Inc., A-2106, Sinsu-ro 767, Suji-gu, Yongin-si 16827, Gyeonggi-do, Republic of Korea
2 Department of Transportation Engineering, Myongji University, Yongin-si 17058, Gyeonggi-do, Republic of Korea
3 Korea Maritime Institute (KMI), Haeyang-ro 301-26, Yeongdo-gu, Busan 49111, Republic of Korea
4 Transportation Research Institute (IMOB), Martelarenlaan 42, 3500 Hasselt, Belgium
5 Korea Railroad Research Institute, Uiwang-si 16105, Gyeonggi-do, Republic of Korea
6 Department of Geography, Kyung Hee University, Seoul 02447, Republic of Korea
* Correspondence: jaeheon.choi@varodrt.com (J.C.); sjcho@kmi.re.kr (S.C.)

**Abstract:** DRT service, designed to be flexible in time and space, follows the contemporary trend of on-demand transit provision. However, this type of service often suffers from low profitability due to small demand and/or high operation costs. DRT service is a local business in nature. The existing research primarily focuses on DRT service for regions with low transit demand, but it does not take into account service operation for other types of regions. This study aims to fill in this gap and identify the sources of benefit from DRT operations in varied types of regions. To this end, the analysis compares the DRT operation performance between overpopulated heartland and underpopulated hinterland regions; in each region, the benefit is identified through the difference in key performance indices between the simulated DRT and actual bus operation. The data on the road network and bus operation in Daegu, Korea, in 2021 are used for the DRT simulation. The results show that the heartland DRT benefits more from the reduced vehicle kilometers, while the hinterland DRT gains mostly from the reduced waiting time. Given that both DRT types outperform existing bus services, it is revealed that the heartland DRT is more reliable than the hinterland DRT due to the nature of regional characteristics.

**Keywords:** DRT (demand-responsive transit) simulation; bus operation; performance indices; heartland DRT; hinterland DRT

## 1. Introduction

On-demand service and the sharing economy have been the dominant trends in contemporary society, particularly in the field of public transport [1]. Cities increasingly tend to introduce demand-responsive transit (DRT) services, in order to overcome the inefficiencies of conventional public transport in terms of operation costs [2–5], delivery management in spacious service areas [6,7], service quality, and user experience [8,9]. Ref. [10] defines DRT as a flexible and intermediate transit mode that fills the gap between individual taxi-type service and scheduled fixed-route conventional transit. The benefit of DRT over existing transit has been reported in many studies, including increasing operation efficiency [2,4,9,11–14], improving user service quality and satisfaction [3,4,6,7,9,15], reducing emissions [13,14,16,17], and promoting modal shift (toward transit) [5,13,18]. It has been discovered that DRT outperforms existing transit in areas with low population density and low travel demand.

However, DRT can also run in areas featuring higher population density and/or greater transit demand if the service operation is tailored to the characteristics of the areas. A DRT operation may tune the service scheduling and operation details based on the call frequency and origin-destination (OD) patterns of the calls, and the performance of DRT can vary with the service details and spatial coverage [3]. Ref. [19] suggests the classification of four distinguished DRT applications. Firstly, the 'interchange DRT' is a feeder to bring people (mostly commuters) to the traditional mainline transit stops and stations such as subway stations and intercity bus terminals. The regional characteristics applied for this type typically belong to the downtown or heartland of a city. Secondly, the 'network DRT' strengthens the transit system by providing additional service at specific times and locations or by replacing the existing, inefficient transit service at non-peak hours and in distant peripheral regions. The regional characteristics for this type mainly typify the peripheral part or hinterland of a city. Thirdly, the 'destination-fixed DRT' is a special case of the 'network DRT' which runs to fixed destinations (such as airports or company locations) from scattered origins. This type is primarily designed for commuter or shuttle service. Fourthly, the 'substituted DRT' is to completely replace the existing transit service.

The regional characteristics, such as transportation infrastructure, building locations, and foot traffic, affect user travel patterns and demand intensity, which in turn determine the role of DRT. For example, the 'network DRT' that replaces the existing bus services of fixed routes and schedules is necessary when transit demand is low and the service area is large. Alternatively, a wide-coverage 'network DRT' is required for the connection between two remote areas. Moreover, the 'interchange DRT' plays the role of feeder transit to deliver passengers from downtown subway stations and major terminals to residential areas. As such, it is very important to adopt the appropriate type of DRT for the specific operating environment.

The existing literature on DRT has mostly been limited to the 'network DRT' type [6,8,20–22], in which simulation studies are performed to examine the potential effect of door-to-door DRT service in replacing the existing bus operation in rural as well as interurban areas with low revenue. The results show that DRT operation outperforms existing transit in areas with low population density and low transit demand. DRT operation provides better performance under the demand of 40 to 60 (passengers) per vehicle [2,8,22].

However, as demand increases, performance decreases [2,7,22], and DRT is frequently provided to meet social welfare policies rather than profit maximization. Nevertheless, in addition to the 'network DRT', there are other types of regions with varied demand sizes and regional characteristics; the appropriate type of DRT operation should thus be selected so that the best service can be provided to the corresponding regions. Furthermore, even a single municipality or local area may have multiple types of regional characteristics within its administration territory, each of which may require a different type of DRT operation. In other words, DRT operations would guarantee the operation's revenue only when it best suits the local demand and regional characteristics.

Given the above literature review, the current study aims to examine the difference in the sources of the benefit from DRT operation when the service operates in different areas featuring varied transit demand patterns and regional characteristics. In particular, unlike the existing literature that extensively examined the rural DRT with small transit demand in a relatively larger service area, the study attempts to reveal the source of the benefit and its relative size for the urban DRT with high transit demand in a relatively smaller service area. To this end, this study establishes relevant research hypotheses and tests the hypotheses by comparing the performance indices between the simulated DRT service and actual bus operation. A metropolitan city in South Korea, accommodating two representative regional types (i.e., the heartland and hinterland) within the urban area, is examined. The simulated DRT operation is tailored to each regional type, and a set of

performance indices are derived from the simulated DRT service as well as from the actual bus operation.

The rest of the paper is organized as follows. Section 2 describes the representative types of regions for DRT operation and proposes research hypotheses. Section 3 develops simulation tools and performance indices, while Section 4 introduces the study area and data. Finally, Section 5 presents the analysis results, and Section 6 concludes the paper.

## 2. Regional Types of DRT Operation and Research Hypotheses

### 2.1. Regional Types of DRT Operation

Two general regional types of DRT operation were described in [10], including:

(1) The regions where the population density and transit demand are high.

(2) The regions where the population density and transit demand are low.

Several studies have been conducted to evaluate the transit fleet size and route plan in relation to indicators such as population density, floor space, hourly transit demand, and travel distance. To quantitatively distinguish regional types, the paper adopts the indicators of passenger-km as in Table 1 ([23]). The passenger-km indicator is the travel distance multiplied by the transit demand that can be served per hour. For example, 20 passenger kilometers means that a passenger travels less than 5 km, given the condition of having 4 passengers on board at each hour. If longer trips are frequently required, a shuttle service should be used. The author of [23] recommends a flexible-type minibus for the case of passenger-km of 10 to 20, which corresponds to the above regional type (2) in the rural area. On the other hand, [23] recommends a semi-fixed or fixed-type transit route for the case of passenger-km higher than 20, which corresponds to the above regional type (1) in the urban area.

**Table 1.** Indicative guidance for fleet size related to demand density.

| Trips per Vehicle Hour × Trip Length [i.e., Passenger-km per Vehicle-h] | Suggested Vehicle Choice |
|---|---|
| Less than 10 | Taxi |
| Between 10 and 20 | Taxi(s) or flexible minibus—choice will depend on availability and relative costs locally |
| Between 20 and 50 | Flexible minibus, with lower degree of route flexibility at the higher end of the range |
| Greater than 50 | Largely fixed route bus, with limited deviations |

Based on the above classification, this study defines two specific types of DRT operations, in response to the varied transit demand and characteristics of the regions. The first type is the hinterland. In this type of region, transit demand is mostly decreasing as the population reduces. Due to the decrease in transit demand, the transit service drops, and problems of unequal service provision among different areas within the region occur. At the same time, the car proportion in the modal split is increasing, making the transit demand even smaller and the problems greater. However, despite the problems, local governments' financial burden gets higher because they have to subsidize transit companies to maintain the current service. Local governments have put a large amount of effort into keeping the regional equality of transit service, but often face the problem of a lack of systematic and effective improvement of service and legal support [24]. The hinterland, or the areas where transit service is weak, finds it increasingly hard to sustain the existing transportation systems and service due to the decreasing population and transit demand.

Solving the problems requires a new transportation policy with a new paradigm that is completely different from the past. The transportation policy of this new paradigm does not rely on the existing transit systems (with fixed routes and operation schedules), but instead it attempts to provide service that is more flexible in time and space to best suit

the passengers' needs. The local governments of South Korea therefore plan to expand DRT service in conjunction with the central government in order to satisfy such diverse user needs. For example, the Ministry of Agriculture operates DRT projects on enhancing transportation in rural areas, while the Ministry of Land and Transportation develops similar projects in remote urban areas. The main goals of such projects are to ensure the mobility of people living in underpopulated areas (i.e., to improve the accessibility of the peripheral regions) and lessen the financial burden on the local governments. The solution to achieve these goals is offered as the establishment of regionally customized demand-responsive transportation service systems [25]. An ill-planned DRT introduction would reduce the effects of the projects and result in inefficient budget use and a waste of administrative resources.

The second type is the heartland. The downtown area of a city mostly experiences active transit use due to high population density, with the transit mode proportion (in the overall modal split patterns) being higher than that in underpopulated areas. Transit is the transport mode with the lowest $CO_2$ emissions and pollution while maintaining high energy efficiency. One of the main goals of transportation policies is to increase the share of transit modes while reducing that of cars and taxis. However, the goal has been hindered by the inconvenience of transit use, due to the fixed stations and stops, routes, and operation schedules of transit systems. Moreover, the introduction of new mobility services such as car sharing and ride sharing offers a variety of personalized services and reduces the merits of public services. A new mode of transportation that is both as convenient as a car and as efficient as mass transit is required. The new service should provide shorter travel times and fewer transfers than the existing transit, while still being able to carry large amounts of passengers, as the transit does.

DRT satisfies the requirements of the new service. DRT was originally introduced to overcome the problems of inefficient transit operations in rural areas with a small population and transit demand. Transit is known to be more efficient than DRT in a populated area with higher transit needs [26]. However, as discussed above, the urban transit system is not able to satisfy all passengers in terms of travel time and travel convenience. For example, some origins and destinations require long travel time and a large number of transfers that reduce user satisfaction. A new type of urban-customized DRT service would be utilized to better handle the demand by grouping users in similar ODs and departure times and providing distinctive service to each group. This type of DRT would supplement the limitations of existing transit service in urban areas where mass demand with similar ODs (e.g., from dense residential areas to downtown) occurs during commute hours, causing problems with unnecessary waiting and congestion because existing transit is unable to group users in similar ODs and departure times. The existing transit services visit all stops regardless of passengers' waiting times and thus cause inefficient driving routes with long distances. The application of the DRT hinterland to such demand is also ineffective and would require too high administrative and operation costs. DRT in the urban region should thus find an efficient way of operation by appropriately designing DRT routes and schedules in accordance with the stops and key nodes of mass mobility. This would lead to the rise of the heartland DRT operation. There has been no prior research that has focused on the distinct source of benefit that the heartland DRT should take. The paper aims to fill in this gap.

### 2.2. Research Hypotheses

Ref. [3] illustrates that the performance of DRT potentially varies with the service details in response to different call frequencies and the levels of availability in time and space, as shown in Figure 1. Their theoretical rationale can be further traced back to [27].

In the current study, the purpose is to examine the difference in the sources of benefit between different types of DRT operations corresponding to varied local demand and regional characteristics, in particular urban DRT. The local demand implies the size of the local transit demand, while the regional characteristics refer to the travel patterns

specific to the city's transit infrastructure (particularly the spatial distributions of ODs of these patterns). Moreover, to evaluate the performance and benefit of DRT service, several indices have been utilized, including the passenger benefit, operator profit, and system efficiency [6,28,29]. The passenger benefit is often represented by the reduced average waiting time and total travel time per passenger. The operator profit is frequently indicated by the reduced total vehicle kilometers and vehicle kilometers per passenger transported [6,7,20,28–30]. Among these measures, this study employs the average waiting time and total vehicle kilometers for evaluating the passenger benefit and operator profit as the key indices of the performance, together with the combination of the two indices for assessing the system efficiency. The total travel time per passenger and vehicle kilometers per passenger transported are not explicitly adopted as the indices because the system efficiency index incorporates these as explained in Section 3.2 of Performance Indices. For each type of region, these indices are compared between the corresponding DRT and existing bus operation.

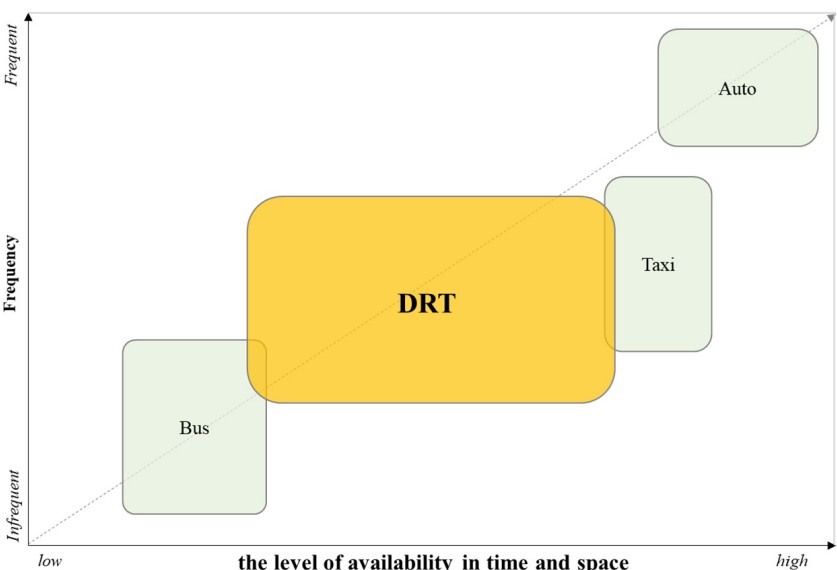

**Figure 1.** DRT in the context of call frequency and the level of availability in time and space.

Based on the above-defined regional types and performance measures, the research hypotheses are designed as follows.

- The background of the research hypothesis: The fundamental characteristics of DRT service lie in the fact that this service utilizes flexible travel routes and operation schedules, replacing the fixed ones in existing transit systems. Different DRT characteristics are reflected in the operation regions that differ in local demand and regional characteristics. Given the basic hypothesis, the research hypotheses that will be tested in this study are as follows.
- Research Hypothesis 1: DRT operation in the hinterland regions gains more benefit from the reduction in average passenger waiting time than from the reduction in total vehicle kilometers, in relation to the existing bus operation.
- Research Hypothesis 2: DRT operation in the heartland regions gains more benefit from the reduction in total vehicle kilometers than from the reduction in average passenger waiting time, in relation to the existing bus operation.
- Research Hypothesis 3: The benefit of the heartland DRT is rather marginal, compared with that of the hinterland DRT.

Figure 2 depicts the workflow of the analysis framework of this study.

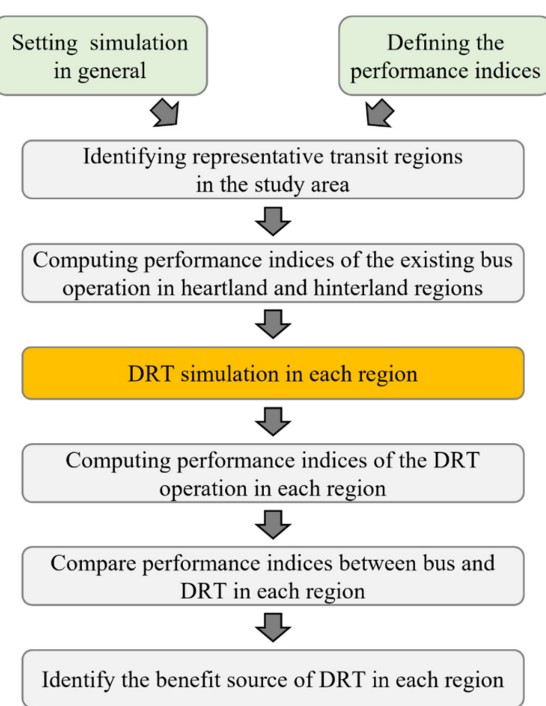

**Figure 2.** Workflow of this study.

## 3. Methodology

### 3.1. Simulation

#### 3.1.1. System Characteristics

The DRT simulation model proposed in this study has two types of agents, including DRT and passenger agents. The simulation serves the passengers' calls for DRT requests during the operation hours. In the course of the service, the indices including passenger travel time and waiting time as well as vehicle driving distance are summed to assess the overall service performance. The interactive actions between passengers and DRT agents are depicted in Figure 3.

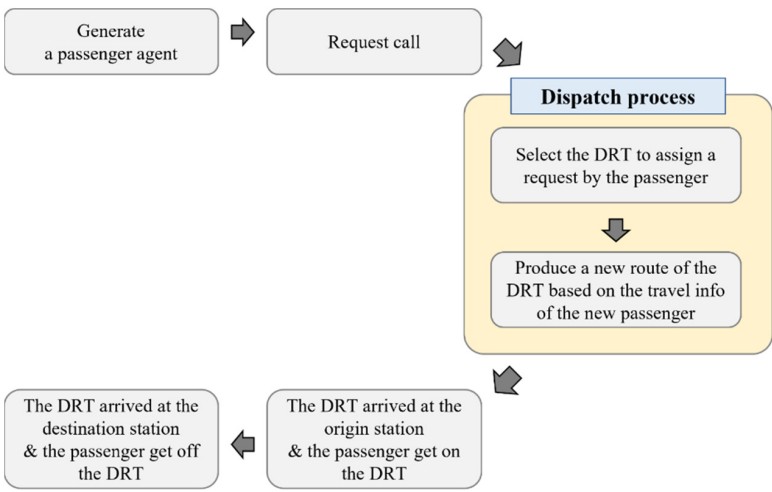

**Figure 3.** The interactive actions between passenger and DRT agents.

The simulation considers two types of DRT including the hinterland and heartland types, and the rules differ between the two. Hinterland DRT is being implemented in areas with lower population density and larger areas. It has many virtual stops and flexible routes between stops. The passengers from hinterland regions mainly use DRT for their daytime travel. Their call times during the day are diverse; while their origins, destinations, and

travel patterns also differ to a high degree. In comparison, heartland DRT is applied to a relatively smaller area with higher population density, in which more calls are requested for the connection to mass transit or intercity transport modes such as subways, airports, and express terminals. This type of DRT often has fixed origins or destinations, and its stops are at local centers. Moreover, heartland DRT primarily serves the demand around particular time slots; for instance, passengers from mass transit may use heartland DRT during commuting hours. As a result, unlike hinterland DRT, heartland DRT can be strategically installed near the ODs of transit passengers to meet the demand of mass transit travelers.

3.1.2. Simulation Model Description

The rules that are common to both DRT types are as follows.

- Passengers should board and disembark at the (virtual) stops, and the vehicle should only stay at the stops. The study area has $N_{stops}$ stops. The interaction between passengers and vehicles solely occurs at the stops.

- Neither passengers nor vehicles can cancel the call. Although in practice, both passengers and vehicles may terminate the call demand, in this simulation, the cancellation of calls is not allowed. This is to be consistent with the conventional bus system which does not provide this cancellation possibility.

The DRT dispatch methods that are different between the hinterland and heartland types are designed as follows. The definition of the used variables can be referred to in Figure 4.

---

$N_{stops}$: Total number of bus stops

$N_{drt}$: Total number of DRT vehicles

$S_{drts}$: Set of DRT vehicles in study area $\{i_1, i_2 \dots i_{N_{drt}}\}$

$T_{drt_i,k}$: Total travel time of the route given to DRT vehicle $i$ at time $k$

$P(T_{drt_i,k})$ : Passenger's $T_{drt_i,k}$

$O_{psg_p}, D_{psg_p}$ : Passenger $p$'s OD

$t_k(node_1, node_2)$ : Travel time between node 1 and 2 at time at time $k$

$n(T_{drt_i,k-1})$: Number of $t_{(node_1, node_2),k}$ in $T_{drt_i,k-1}$

$Loc_{now}(drt_i)$: Current location of DRT vehicle $i$

$Loc_{k-1}(drt_i)$: Location of DRT vehicle $i$ at time $k-1$

$ddist_{r,psg_p}$: Maximum allowed detour distance of passenger p when base detour rate is $r$

$d_{direct}(node_1, node_2)$ : Minimum distance between node1 and node2

$S_{stops}$: Set of stops in study area $\{s_{initial}, s_1 \dots s_{N_{stops}-1}\}$

$s_{initial}$: Initial location of the stop for heartland DRT

$d(s_i, s_j)$ : travel distance between stops $s_i$ and $s_j$

$s_{i-th}$: Order of the stops

---

**Figure 4.** Variable definition.

1.   Hinterland DRT

(1) Dispatch
Because of the need to cover the wide spatial range of the service area, two vehicles are in operation ($N_{drt} = 2$). The rule to determine which vehicle is to be arranged for the passenger is required. Equation (1) explains that vehicle $i$ ($i$ = 1, 2) is arranged such that the total vehicle driving distance between the origin and destination is minimized for the newly served passenger $psg_p$.

$$\operatorname*{argMin}_{i \in S_{drt}} T_{drt_i,k} = f\left(T_{drt_i, k-1}, t_{add_{drt_i}}\left(O_{psg_p}, D_{psg_p}\right)\right) \tag{1}$$

where $T_{drt_i,k}$ is the travel time when the new route is generated at time $k$ due to the addition of the origin and destination to the concerned route during the operation of vehicle $i$ at time $k - 1$.

The dispatch is assigned to the vehicle of the smallest $T_{drt_i,k}$ among the vehicles currently in operation $S_{drt}$. The following is a procedure to compute $f(T_{drt_i,\,k-1},\,t_{add_{drt_i}}(O_{psg_p},\,D_{psg_p}))$ of Equation (1), the following equations are derived.

- When $T_{drt_i,\,k-1} = 0$ (i.e., the vehicle is empty),

$$f(T_{drt_i,\,k-1},\,t_{add_{drt_i}}\left(O_{psg_p},\,D_{psg_p}\right)) = t_k\left(Loc_{now}(drt_i)\,,\,O_{psg_p}\right) + t_k\left(O_{psg_p}\,,\,D_{psg_p}\right) \quad (2)$$

Equation (2) is the sum of the travel time from the current empty vehicle location to the stop where the passenger is waiting and the travel time from boarding (at the current stop) to disembarking (at the destination of the trip).

- When $T_{drt_i,\,k-1} \neq 0$ and $n(T_{drt_i,\,k-1}) < 7$, Algorithm 1 is applied.

---

**Algorithm 1.** An ordinary case when # passengers is 0 to 7

---

01 **Step 0. Initialization**
02 bestPath = None
03 bestPathTravelTime = Integer.MAX_VALUE
04 wayPointsSet = [ $O_{psg_{p1}}$, $O_{psg_{p2}}$, $D_{psg_{p1}}$ ... ] {the list of stops for DRT i to pass by at time k − 1}
05 wayPointsSet = [wayPointsSet $\cup$ $O_{psg_p}$ $\cup$ $D_{psg_p}$ ] {addition of OD of newly assigned passenger p}
06
07 **Step 1. Generating all routes combinations**
08 numOfWayPoints = len(wayPointsSet) {store the number of stopover stops}
09 {store all permutation of orders that can be the list of stopovers}
10 combinationsSet = [comb for comb in combinations(wayPointsSet, numOfWayPoints)]
11
12 **Step 2. Checking whether the optimum routes selects**
13 for comb $\in$ combinationsSet:
14 tempTravelTime = 0
15 removeFlag = False
16 For idx: I $\in$ (0,len(comb) − 1)
17 tempTravelTime += $t_{(comb[idx],comb[idx+1]),k}$ {sum of the travel times between stops}
18 if Criteria1: {if not reversed boarding and disembarking, then True}
19 removeFlag = True
20 break
21 End if
22 if Criterial2: {if passengers exceeds the capacity, then True}
23 removeFlag = True
24 break
25 End if
26 if Criteria3: {if passenger's maximum detour ratio exceeds, then True}
27 removeFlag = True
28 break
29 Endif
30 End for
31 if removeFlag == False {if routes passes all the criteria, then False}
32 if bestPathTravelTime > tempTravelTime:
33 bestPathTravelTime = tempTravelTime
34 bestPath = comb
35 End if
36 End if
37 End for
38
39 **Step 3. Optimum vehicle route and total travel time**
40 $T_{drt_i,k}$ = bestPathTravelTime
41 P($T_{drt_i,k}$) = bestPath
42 End

---

- When $T_{drt_i, k-1} \neq 0$ and if $n(T_{drt_i, k-1}) \geq 7$,

$$f(T_{drt_i, k-1}, t_{add_{drt_i}}(O_{psg_p}, D_{psg_p})) = T_{drt_i,k-1} + t_k(Loc_{k-1}(drt_i), O_{psg_p}) + t_k(O_{psg_j}, D_{psg_p}) \quad (3)$$

$n(T_{drt_i, k-1}) \geq 7$ would need too much computing power of more than 9! (=362,800) route sets to search including $O_{psg_p}$ and $D_{psg_p}$. Once this initial search of optimal schedule is done, Equation (3) is applied to a simple addition of the new passenger's origin and destination to the optimal route already found by Algorithm 1 in the previous step.

(2) Detour rates

The above dispatch method, which minimizes the DRT travel time for each call, may cause long delays for a passenger. This violates the passenger's punctuality which is the basic characteristic of transit. Therefore, a maximum detour rate *r* is set to not allow passengers to detour more than a threshold.

$$ddist_{r, psg_p} = d_{direct}(O_{psg_p}, D_{psg_p}) * (1 + r) \quad (4)$$

For example, when *r* is 1.0, DRT should deliver the passenger *p* within twice the travel time required for the shortest distance from the boarding to the disembarking.

2. Heartland DRT

(1) Dispatch

The initial stop location is $S_{initial}$, which may be a subway station or a location farthest from the subway depending on the time of the day and passenger travel patterns. Heartland service is installed to serve the time-specific demand. DRT route is directional, with the direction starts from $S_{initial}$. The order of stops along the route is as follows.

$$s_{ith} = \min d(s_{(i-1)\,th}, s_{ith}) \quad (i = 1,2,3 \ldots (N_{stops} - 1); \; s_{0th} = s_{initial} \quad (5)$$

$$s.t.d(s_{initial}, s_{(i-1)\,th}) < d(s_{initial}, s_{(i)\,th}) \quad (6)$$

According to the order in Equation (5), the vehicle travels as follows.

$$s_{initial} \rightarrow s_{1th} \rightarrow s_{2th} \rightarrow \cdots \rightarrow s_{(N_{stops}-1)th}$$

Opposite travel order (such as $s_{5th} \rightarrow s_{2th}$) is not allowed.

It is important to note that even if the stop is on the route, it is skipped when no call is received. Only the stops with a call at the moment of the drive are approached by the vehicle along the shortest route. This makes the suggested DRT different from the existing bus services in the urban region. The vehicle goes back to the initial location when it becomes empty. For example, when the vehicle at $s_{1th}$ receives calls from $s_{7th}$ and $s_{4th}$ at 3:12 p.m. and 3:14 p.m., respectively, the vehicle route is generated as follows.

$$s_{1th} \rightarrow s_{4th} \rightarrow s_{7th} \rightarrow s_{initial} \text{ (back to origin)}$$

### 3.1.3. The Simulation Tool

This study uses netlogo for simulation. Figure 5 shows the netlogo interface. This tool accommodates a number of functionalities, consisting of the simulation GUI, travel time and distance calculation, scenario setup, program initialization and run, and indicators dashboard, as well as various parameter settings, including the demand assignment, DRT fleet size, simulation execution parameter, maximum waiting time, maximum detour rate, and DRT capacity.

### 3.2. Performance Indices

A number of performance indices have been developed to evaluate transit operations [6,28,29]. Table 2 shows the ones that are broadly used for evaluating transportation services and infrastructure.

Performance indices for passenger benefit reflect the degree of disutility when passengers use the transit. Both longer waiting time and total travel time per passenger contribute to the increment of disutility. Meanwhile, indices for operator profits indicate the operation costs, both longer vehicle kilometers per passenger transported and longer total vehicle kilometers increase the costs. A combination of the indices (i.e., TUC) from both the passenger and operator perspectives determines the system performance representing the overall performance of the transit. TUC converts

both user-experiencing service level and operation performance into monetary value according to Equation (7).

$$TUC = \frac{\sum APTT * VOT + OC}{Number\ of\ calls\ served} \tag{7}$$

where *VOT* is value of time in money, *OC* is operation costs in money.

**Table 2.** Performance indices used in the transportation analysis.

| Performance Index | Evaluation Target | Description |
|---|---|---|
| AWT (s) | Passenger | Average waiting time |
| APTT (s) | | Average passenger travel time |
| TI (km) | Operator | Transport intensity |
| TDD (km) | | Total driving distances |
| TUC | System | Total unit costs per passenger |

Norte: AWT is measured as the average interval for bus and the average waiting time from call to arrival for DRT. APTT is measured as the total travel time (waiting time + in-vehicle time) per passenger. TI is measured as the vehicle kilometers per passenger transported. TDD is measured as the total vehicle kilometers. TUC is measured as in Equation (7) below.

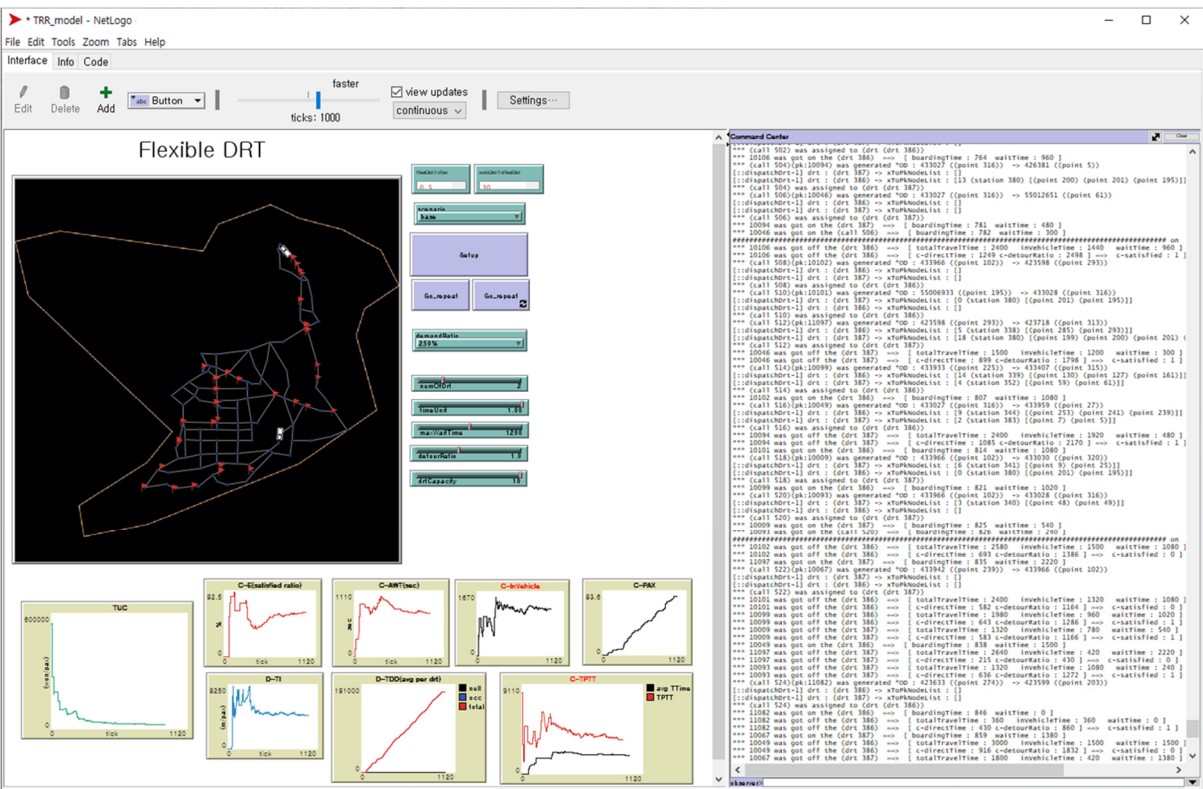

**Figure 5.** The simulation user interface.

TUC includes the conversion of the VOT of user-experienced service travel time (waiting time + in-vehicle time) into costs and the operation costs such as vehicle running costs per day, fuel costs, driver payroll costs, vehicle depreciation costs, insurance costs, and repair and maintenance costs. The existing DRT literature using TUC [6,28,29] applies the concept of weighted average travel time, which assigns a high travel time to passengers who fail to experience satisfaction because the level of service is below some minimum threshold. However, in this study, APTT is adopted without such weight because the satisfactions of DRT and bus are not estimated to be comparable, and therefore TUC does not concern satisfaction.

VOT of the current bus mode in Korea is assessed as KRW 13,440 (or approx. USD 9) [31]. OC is the sum of fuel and transportation costs (excluding fuel costs) per vehicle. Fuel costs are computed per liter multiplied by TDD against gas mileage. According to the Standard Transport Costs [32],

transportation costs, excluding fuel, are assessed at KRW 512,739 (or approximately USD 357) per vehicle per day.

As stated in Section 2.2, this study includes AWT, TDD, and TUC as the key performance indices. APTT and TI are not listed. Without the call cancellation, TI is proportionally the same as TDD in this study. APTT is explicitly included in the calculation of TUC, and AWT is further developed, together with detour rate, to examine the reliability of different DRTs in the later stage of this study.

## 4. Study Area and the Data

### 4.1. Study Areas

The two areas' distinct characteristics are appropriate for comparing DRT performance between places with differing regional characteristics and travel behaviors of their inhabitants. It is a city of urban–rural integration, consisting of seven administrative 'Gu' of urban regions and one 'Gun' of rural region, as shown in Figure 6.

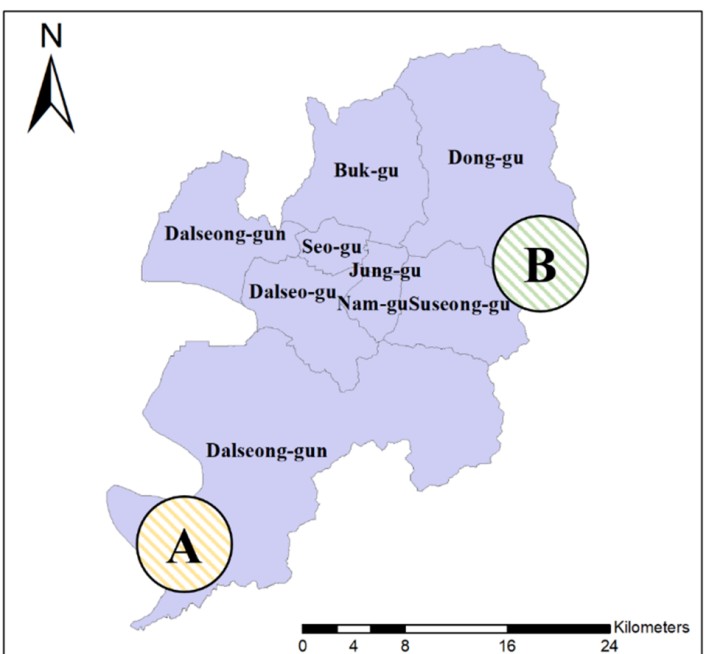

**Figure 6.** The city of Daegu.

The two study areas (including Area A and Area B) within the city are selected to compare the simulated DRT and the existing bus services, as the two areas are representative examples of an infrequent and inefficient rural bus line and a busy urban feeder bus line, respectively. Area A has a rural environment, whereas Area B features urban settings. Specifically, in Area A, the transit service is vulnerable because only one bus line connects with neighboring subcenters [33]. In contrast, in Area B, the businesses, institutions, and schools are densely located, and a large travel demand is created during commute hours due to the regular commute travel patterns that typically occur in large cities. The distinct characteristics of the two areas are appropriate for comparing DRT performance between places with varying regional characteristics and travel behaviors of their inhabitants.

This study designs two types of DRT to simulate the operation and evaluate the performance of DRT in relation to the performance of the existing transit in those regions. The details of Area A are presented in Figure 7. The simulation installs the hinterland DRT operation in this area and compares the performance indices with those of the existing bus operation. The administrative name of the area is Guji-Myeon, Dalsung-Gun, Daegu. The bus line connects the low population density area and the local center of a new town, called Daegu Techno-polis, in the area. Currently, there is only one bus running along the line, 5 to 6 times a day. The total length of the line is 53 km, and the total driving time is 2 h and 20 min for a return trip, while the dispatch interval is 3 h. The demand for bus transit is very low. The consequence of the regional characteristics and low demand is to very long waiting times, while the fixed route keeps the total vehicle kilometers constant.

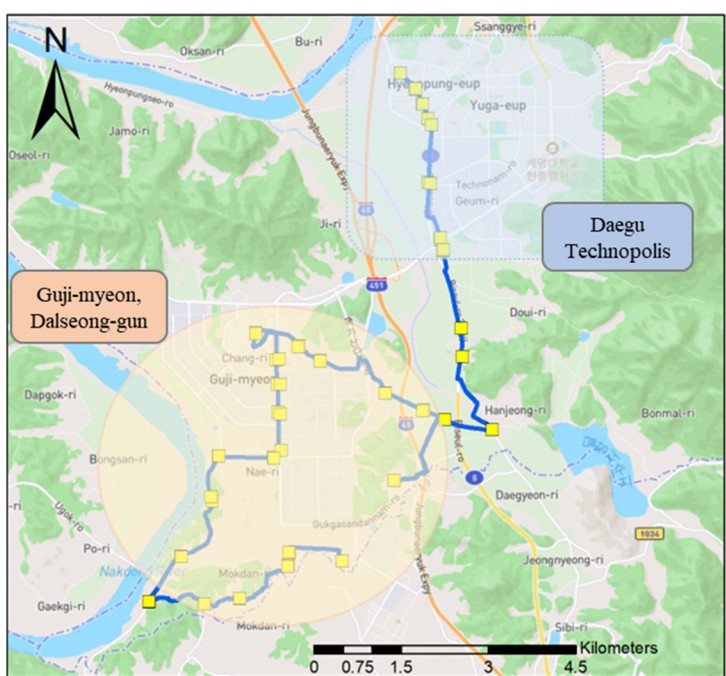

**Figure 7.** Bus route in the rural area in Daegu.

The hinterland DRT route in the simulation is designed for the inhabitants in the rural periphery to easily access the local center by means of the DRT service. The DRT operation catches the call at any time and place along the route, in order to provide services that are more flexible in time and space to suit the passengers' needs. The simulated DRT service also runs along the bus line, and the performance is compared to that of the current bus operation.

The details of Area B are depicted in Figure 8. The simulation sets up the heartland DRT operation in this area and compares its performance with that of the existing bus operation. The administrative name of the area is Dong-gu, Daegu, which features a high population density and high travel demand. The bus line connects the subway station and the Research Complex, called Innovation City. The research complex is equipped with an innovative industrial research environment and settlement, enabling close collaboration between companies, universities, and public institutes. The bus line serves as an intermodal mode to connect the research complex and subway station for commuting and schooling. The total length of the bus line is 9.1 km with 8 stops in between, and the line currently has one bus running with 15 min dispatch intervals. The regional characteristics and high demand typically cause the fixed total vehicle kilometers during the peak hours because the fixed route stops at every stop regardless of the existence of passengers waiting.

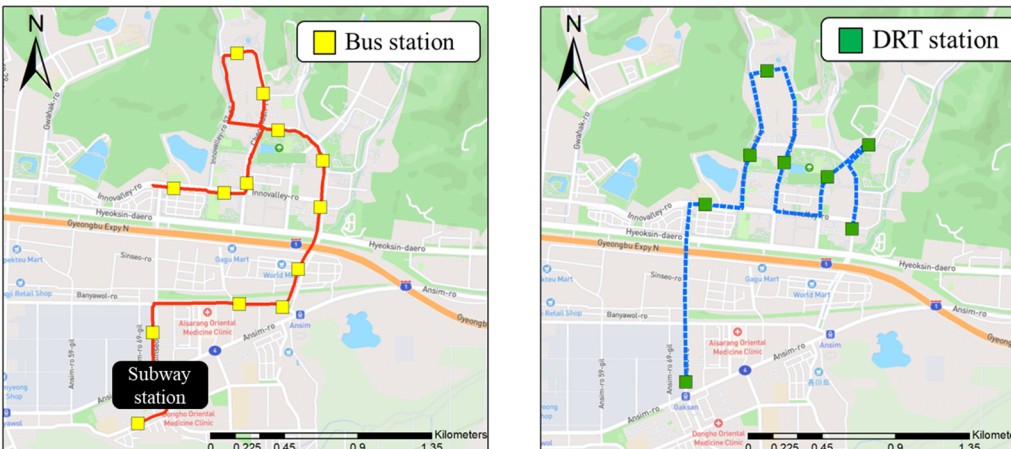

**Figure 8.** Bus route (**left**) and a DRT path in response to the real-time call (**right**).

The heartland DRT route is designed to better serve the passengers along the line at peak hours of the day: 07:30–10:00 a.m. and 17:30–20:00 p.m. The DRT simulation runs from the research complex to the subway station during the morning peak and from the subway station to the research complex during the evening peak. The DRT vehicle stays near the subway station in the morning peak and the uppermost waiting location in the evening peak because all passengers then disembark, and the vehicle can catch the call faster. The simulated heartland DRT service also runs along the bus line, and the results are compared with those of the current bus operation regarding the performance indices.

### 4.2. The Data

### 4.2.1. Travel Pattern

(1)     Hinterland bus services

The data on the demand for the route in the region are retrieved from the smart card to determine the size of the demand for the existing bus line 7 currently in operation. The data were collected on Tuesday, 15 October 2019, to avoid the seasonal effect on travel demand. On that day, 76 riding cases of travel demand were discovered. The travel pattern is shown in Figure 9.

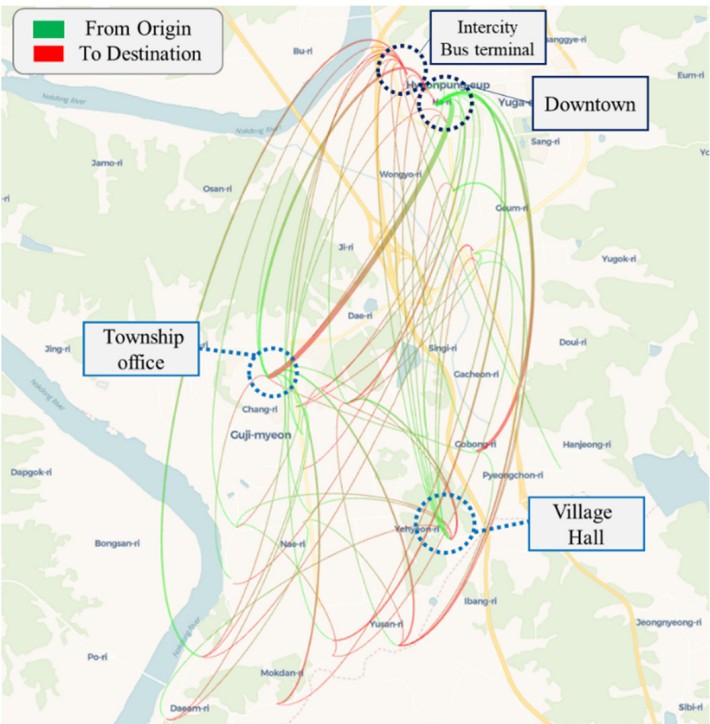

**Figure 9.** Travel pattern of the existing bus line 7 in Dalsung.

The travel occurs mostly between the local rural area and the suburban new town area called Daegu Technopolis, rather than within the local rural area. The average time spent in the vehicle is 1895 s, or approximately 31 min. The average waiting time of 90 min was inferred from the smart card as being half of the dispatch interval of 180 min.

(2)     Heartland bus services

Travel occurring in the study area is directional, and bus line 4-1 travels mostly from the subway station to the local bus stops, and line 4 serves travel from local bus stops to the subway station. The data on commutes to and from the industrial complex for a week, from 9 May 2022 of Monday to 13 May 2022 of Friday, were collected and used for the analysis. The travel patterns of the morning peak and evening peak are compared with each other, as shown in Figure 10. Most travel involves subway stations as the origin or destination and travel other than to the subway station is rare. In total, 41 riding cases were identified. The average in-vehicle time is 428 s, or approximately 7 min. The average waiting time was inferred as 464 s, or approx. 7 min.

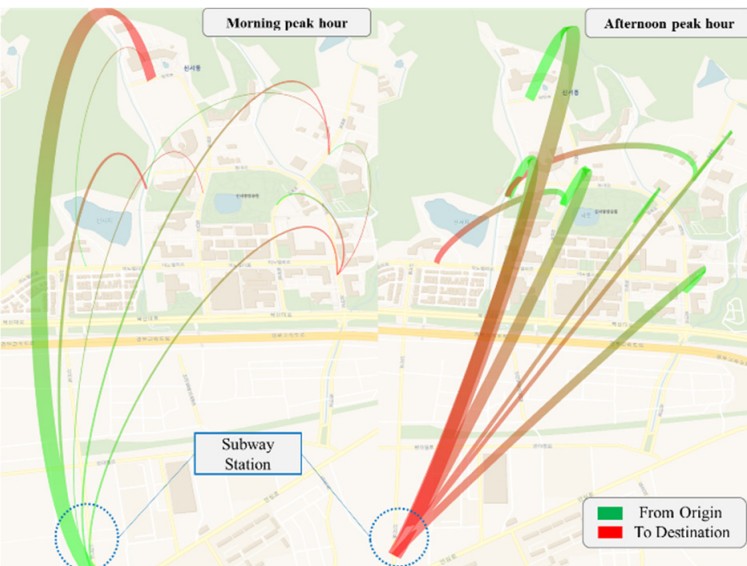

**Figure 10.** Travel pattern of the existing bus lines 4-1 (morning to the Complex) and 4 (evening to the subway station) in Dong-gu.

### 4.2.2. Input Data Generation

As for the hinterland DRT, the input utilizes the bus card data of Dalsung bus line 7, which provide information on the number of passengers riding. Boarding time depends on the bus schedule on the run, and the boarding time inference provides randomness of $\pm$ 90 min given the information of a dispatch interval of 180 min; 90 min is the average expected waiting time. As for the heartland DRT, the expected demand is 41 calls during the peak hours (morning and evening) of a day, sampling from the historical data of a 1-day smart card. The heartland DRT collects the data from morning and evening commuters which are less flexible throughout the day. The two data sets consist of 30 repetitions for simulation.

### 4.2.3. Simulation Setting

(1)   Hinterland DRT

The setting for the simulation of hinterland DRT is the following: operating hours are from 06:50 to 22:10 for the existing bus service. The number of vehicles is 2, and the bus capacity is 12 people. The number of virtual stops is 57, which is the same as the existing Dalsung bus line 7. The route is flexible, and the passenger may call any of the 57 virtual stops. When receiving the call, the DRT vehicle goes to the passenger via the shortest path. The number of Dalsung buses is also assumed to be two.

DRT can transport more passengers than a taxi but fewer than a bus. Passengers delivering more than a certain threshold can cause the inefficiency of the DRT operation, which has been proven by [1,17,30,34,35]. It is therefore highly important to introduce the appropriate number of DRT vehicles. The authors of [22] report that, depending on the service operation range, a DRT vehicle serves 4 passengers per hour, whereas a bus vehicle serves 12 passengers per hour. The analysis confirmed that 28.6 DRT vehicles are required to handle 1020 calls, which amounts to 36 calls per vehicle. This study assessed that two DRT vehicles are required for the hinterland DRT operation to handle 76 calls a day.

(2)   Heartland DRT

Operating hours are 07:30 to 10:10 in the morning peak and 17:30 to 20:00 in the evening peak. The number of vehicles is 1, and the bus capacity is 12 people. The number of virtual stops is 8, which is the same as the existing Dong-gu bus lines 4-1 and 4. The passenger may call any of the 8 virtual stops. That DRT comes to a halt without a call, and no disembarking occurs. When receiving the call, the DRT vehicle goes to the passenger via the shortest path. When the passenger disembarks and it becomes empty, the vehicle goes to the waiting points at the central location and the farthest locations.

## 5. Result Analysis

### 5.1. Hinterland DRT and the Bus

The biggest merit of DRT introduction is the reduction of AWT shown in Table 3. In the hinterland DRT case, the bus waiting time is conceptually the same as the loss time. The passengers of the existing bus service would know the bus schedule due to the very long interval. Therefore, they do not 'wait' but they must spend time adjusting themselves to the inconveniently long waiting time. DRT removes this problem, as it can be called at any time and space when needed.

**Table 3.** Performance indices of the existing bus operation and *hinterland* DRT simulation in Area A.

| Performance Indices | Bus | DRT | | +/− (%) |
|---|---|---|---|---|
| TUC | KRW 34,611 | KRW 21,318 | (1236.0) | −38.4% |
| AWT (s) | 5400 | 709 | (150.0) | −86.9% |
| TDD (km) | 292.8 | 319.0 | (23.6) | +8.9% |

Note: # DRT vehicles is 2; # passengers is 76; # iterations is 30; parentheses are the standard deviation.

The existing bus guarantees punctuality because it runs at a fixed time and on a fixed route. However, DRT changes the route in direct response to a call, and the level of service for the passenger (in-vehicle time and waiting time) is greatly affected by the intensity of calls and the locations of start and end points of the vehicle. This may affect the punctuality or the reliability that the DRT passenger.

To guarantee the promised level of service, it is often the case that the call from the passenger is canceled by the operator when the call is far away from the current vehicle location, or the route is generated under the detour rate threshold within the maximum extra in-vehicle travel time of the passenger. Here, detour rate means the extra time computed as the difference between the direct shortest distance and actual DRT vehicle distance. However, this study did not apply the maximum waiting limit and detour rate ceiling because the study needs to compare the performance indices of the DRT operation with that of the existing bus services.

Table 4 states that 50% of the service users had a waiting time shorter than 10 min, and 80% had a maximum of 17 min. The service is designed to allow a maximum waiting time of 40 min. A detour takes place when collecting a new passenger or dropping the current onboard passenger, which delays the vehicle operation. For example, when a direct trip from boarding to disembarking took 10 min while DRT took 16 min, the detour rate is computed as 60% = (16 − 10)/10. Half of the passengers experienced detours of 55% or less, and 20 to 30% of the passengers took time twice as long as the direct travel. The table indicates that a passenger normally taking 10 min by taxi from onboarding to disembarking places needs to call 41 min before the schedule to arrive in time with 80% of success probability; 41 min is computed as the sum of the waiting time of 17 min plus in-vehicle travel time of 10 min + 10 min × 140% = 24 min.

**Table 4.** Area A's DRT service reliability (waiting time and detour rate).

| Attribute/Percentile | 50% | 60% | 70% | 80% | 90% | 100% |
|---|---|---|---|---|---|---|
| waiting time (min) | 10 | 12 | 14 | 17 | 23 | 40 |
| detour length against in-vehicle travel time (%) | 55% | 66% | 88% | 140% | 260% | 587% |

### 5.2. Heartland DRT and the Bus

Heartland DRT achieves improvement in both passenger and operator indices, compared with the existing bus services (Table 5). This improvement can be originated from the following two aspects. Firstly, DRT skips the virtual stops not in the boarding or disembarking schedule at the moment of the operation. The vehicle, therefore, travels around only at the called stops along the optimal route. This decreases vehicle operation distance and passenger travel time. Secondly, the DRT operation in this region deals with a very simple demand pattern focused on a specific time window. That is, the morning peak is observed mostly from the subway station to the local virtual stops of the Complex, while the evening peak witnesses the opposite. The empty vehicle staying back to the original starting virtual stop contributes to the reduction in waiting time.

**Table 5.** Performance indices of the existing bus operation and heartland DRT simulation in Area B.

| Performance Indices | Bus | DRT | | +/− (%) |
|---|---|---|---|---|
| TUC | KRW 12,513 | KRW 11,633 | (429.0) | −7.0% |
| AWT (s) | 464 | 388 | (54.3) | −16.4% |
| TDD (km) | 107 | 62.7 | (3.6) | −37.7% |

Note: # DRT vehicles is 1; # passengers is 41; # iterations is 30; parentheses are the standard deviation.

Table 6 states that Area B's waiting time and detour rate were much smaller than Area A because the fixed type of vehicle operation covers a much smaller area. This implies higher mode reliability. The table explains that a passenger commuting from the subway station in the rush hour to the workplace in 8 min by car needs to call a DRT 22.64 min before the subway station for arrival at the entry of the workplace with an 80% probability of success; 22.65 min was computed as a waiting time of 10 min + in-vehicle time of 8 min × 1.83.

**Table 6.** Area B's DRT service reliability (waiting time and detour rate).

| Attribute/Percentile | 50% | 60% | 70% | 80% | 90% | 100% |
|---|---|---|---|---|---|---|
| waiting time (min) | 7 | 7 | 8 | 10 | 12 | 18 |
| detour length against in-vehicle travel time (%) | 25% | 29% | 71% | 83% | 100% | 180% |

### 5.3. Hinterland and Heartland DRTs

This subsection summarizes a direct comparison of hinterland and heartland DRTs from the performance indices of Tables 3 and 5 and the reliability presented by Tables 4 and 6.

#### 5.3.1. Performance Indices of Hinterland and Heartland DRTs

Overall, the service quality improvement by DRT introduction is much higher for the hinterland region than for the heartland region because the quality of transit services has been much poorer in the hinterland region than in the heartland region. The hinterland TUC improvement of 38.4% is much bigger than the heartland TUC improvement of 7%. Hinterland DRT particularly much improves the index of AWT by −86%, whereas having a negative impact on the index of TDD at +8.9%. Heartland DRT primarily improves the vehicle travel distance of TDD by −37.7%, while the improvement of waiting time (AWT) is moderate at −16.7%. In other words, hinterland DRT has a strong effect on waiting time improvement, whereas heartland DRT's effect is characterized by vehicle travel distance.

#### 5.3.2. Reliabilities of Hinterland and Heartland DRTs

The level of reliability is affected by whether the major travel patterns of the region are focused on a particular time and space. Heartland DRT of this study has unique spatial characteristics of mass transit stops and stations and clustered residential areas together with the temporal concentration of commuting in the morning and evening. These unique travel patterns result in directional movement to and from the mass transit stops and stations in the morning and the evening, respectively. This is often the case in the heartland areas concerning the commuting trip. On the other hand, hinterland traffic is particularly scattered in time, and also the residential locations in the larger area are also relatively scattered than concentrated.

The vehicle travel distance per passenger of hinterland DRT is much longer than that of heartland DRT. The former is 4.20 km (=319.0 km/76 passengers), whereas the latter is 2.09 km (=62.7 km/30 passengers). This difference affects the difference in average waiting time and its variance between the two DRTs. The median waiting time is 10 min and 7 min for hinterland and heartland DRTs, but 90% of the waiting time is 23 min and 12 min, respectively, which indicates a much bigger variance for hinterland DRTs. The median detour rate is 55% and 25% for hinterland and heartland DRTs, and 90% for detour rate is 60% and 100%, respectively, which also indicates much lower variance inducing the lower variability or higher reliability of heartland DRT.

*5.4. Discussions*

The summary of the research findings is as follows, based on which the decisions on the research hypothesis of this study are drawn.

(1) Hinterland DRT benefits from the reduction in waiting time, whereas heartland DRT obtains the benefit from the reduction in total vehicle operation distance.

(2) Hinterland DRT's benefit is high while heartland DRT's benefit is marginal, caused by the different nature of the applied regional types.

Given the above findings, Research Hypotheses 1, 2, and 3 are therefore all accepted.

The study would summarize the above simulation analyses as follows (Figure 11). First, hinterland DRT gains the benefit of saving time by promptly responding to even the infrequent calls that were not able to be reached on time by the existing bus service. The benefit is therefore primarily the reduction in passenger waiting time. Second, heartland DRT gains the benefit of optimizing the route by skipping the stops that had no call demand at the moment of the operation but were unnecessarily visited by the existing bus service. The benefit is therefore primarily the reduction in vehicle kilometers.

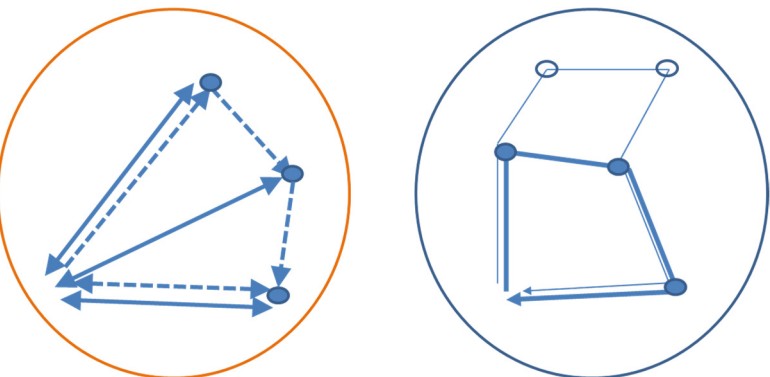

**Figure 11.** Different DRT effects: hinterland DRT effect (**left**) and heartland DRT effect (**right**).

Hinterland DRT: In the case of the existing bus, the service is served as scheduled regardless of when the demand occurs, whereas DRT service is provided rightly on demand.

Heartland DRT: The existing bus service operates the vehicle as routed even if the bus stops do not have demand at the moment of the operation, whereas DRT service is served only the bus stops having demand at the moment of the operation.

As a result, this study can clearly claim that the source of benefit of the urban DRT differs from that of the rural DRT and derives an important implication of the need for choosing an appropriate DRT type for different regional characteristics, which has never been explicitly proven by the comparison between the simulated DRT and the bus services currently in operation in the previous DRT studies of simulation. The level of reliability is affected by whether the major travel patterns of the region are focused on particular times and spaces. The reliability regarding waiting time and detour rate, the heartland DRT is shown to be higher than the rural DRT given the nature of the service area regarding the intensity of transit demand in time and space.

## 6. Conclusions

The purpose of this research is to identify the sources of benefit from DRT operations in different regions. To this end, the analysis compares the DRT operation performance between overpopulated and underpopulated areas. The hypotheses that the sources of benefit differ depending on regional characteristics are tested. Moreover, this study defines the DRT operating in rural areas as the hinterland DRT and the one in urban areas as the heartland DRT. The data on the road network and bus operation of Daegu, Korea, in 2021 are used for simulating the DRT operation, and the results are compared with real bus use data.

The main findings are as follows. Firstly, the hinterland DRT outperforms the existing bus transit service by decreasing passenger waiting time. This finding confirms the acceptance of Research Hypothesis 1 which states that the hinterland DRT operation gains more benefit from the reduction in average passenger waiting time. Secondly, the heartland DRT outperforms the existing bus transit service by reducing vehicle kilometers. This finding verifies the acceptance of Research Hypothesis 2

which assumes that the heartland DRT operation gains more benefit from the reduction in total vehicle kilometers. Thirdly, Hinterland DRT's benefit is high while heartland DRT's benefit is marginal, caused by the different nature of the applied regional types. This finding proves the acceptance of Research Hypothesis 3 which the benefit of the heartland DRT is rather marginal, compared with that of the hinterland DRT. In addition to the size of the benefit, the reliability regarding waiting time and detour rate, the urban DRT is shown to be stronger than the rural DRT given the nature of the service area with regard to the intensity of transit demand in time and space.

The implication of the above-mentioned results is paramount. Firstly, the results convey a clear contribution to the academic field. Unlike most existing research on DRT simulation, where the best combination of demand and fleet size is sought given the type of DRT service, an integrated analysis framework is suggested in this study to examine the performance of the newly introduced DRT service in comparison with the existing transit service in multiple regions of different characteristics. The distinctive nature of DRT operation is its flexibility in time and space, following the passenger needs as they change over time and the on-site optimization of vehicle routes. The two core components of this nature, time and space, appear to vary with regional characteristics, in particular, the population density in the service area. The analysis clearly shows that the strength of the DRT operation in comparison with the existing transit service differs between service areas with different regional characteristics. Secondly, the results also offer practical use for business opportunities and support local governments' decisions on the type of DRT service to provide. Particularly, the operating costs can be reduced, and the passenger demand can be increased by appropriately using the information provided by the current analysis, such that the underpopulated area would arrange both flexible routes and flexible intervals, whereas the overpopulated area would have only marginal variations of the existing fixed routes and fixed intervals depending on the time and space of the real-time call. The primary contribution of this study is summarized as follows: the different sources of benefit of DRT operation among different regional characteristics were first ever examined by a comparison between the simulated DRT and the bus services currently in operation.

A further research agenda is identified as follows. First, other types of DRT operations are required to be tested for more general evidence. Following [19], there are many other types of DRT operations possible. More specifically, it must often be the case that the destination-fixed DRT operation is required. Second, the improvement of the DRT service in the hinterland and the heartland regions should be more clearly explained, or visualized, for the real-time DSS (decision support system) to easily improve the diverse DRT service. Finally, this study identifies the performance improvement of the simulated DRT operation in comparison with the existing bus services in actual operation regarding a simple set of crucial indices. The systematic relationships of such an improvement of the key indices with the external indices potentially affecting the DRT performance, such as population density, the operation plan, and the floor space of the service area, should be investigated further, as this will provide important information for the detailed planning of the DRT operation.

**Author Contributions:** Conceptualization, H.K., H.J. and J.L.; data collection, J.C.; methodology, H.K. and S.C.; formal analysis, J.C., S.L. and D.K.; writing—original draft preparation, J.C. and S.C.; writing—review and editing, F.L. and C.-H.J. All authors have read and agreed to the published version of the manuscript.

**Funding:** This work was supported by the Korea Agency for Infrastructure Technology Advancement (KAIA) grant funded by the Ministry of Land, Infrastructure and Transport (No. 21NSPS-B149565-04) and (No. RS-2022-00143647).

**Informed Consent Statement:** Not applicable.

**Data Availability Statement:** No data to report.

**Acknowledgments:** The authors greatly appreciate the support of the Korea Railroad Research Institute.

**Conflicts of Interest:** The authors declare no conflict of interest.

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
