# Peer review of "Identifying Different Sources of the Benefit: Simulation of DRT Operation in the Heartland and Hinterland Regions"

_sustainability, doi:10.3390/su142416519_

Round 1
Reviewer 1 Report
In attachment

Author Response
Dear Reviewer,
This study aims to fill in the gap and identify the sources of the benefit from DRT operation in varied types of regions. The analysis compares the DRT operation performance between over-populated and underpopulated regions in order to test the hypothesis that the sources of the benefit vary with the regional characteristics.
The analysis results show that the benefit from DRT operation is identified differently in each region, where the key performance indices of the simulated DRT operation were computed against the actual bus operation. The results imply that the policy focus of the DRT operation should differ between over-populated and underpopulated regions to maximize the benefit from the DRT operation.
*We have summarized a point-by-point response to your comments in the attachment. I would appreciate it if you could check the attachment.

Reviewer 2 Report
The article takes the benifit of the new public transport operation model DRT as the research object. The topic and content sound novel, but the organization and writing of the manuscript is not rigorous enough, and the indicators are too simple.
1. In the introduction, the authors tried to explaine the practical problems of DRT but did not clearly point out what practical problems this article should deal with and solve.
2. Although the authors have quoted many literatures, they have never discussed the research question to be solved in this study, and the research object and content are not clear.
3. The purpose of the research is unknown. How can the research results guide the practice? Are there any important findings? The conclusions stated by the authors can be simply drawn without simulation and calculation just through intuitive judgment.
4. What are the similarities and differences between the existing research in the aspacts of research ideas, research methods and research results? The research gap is not explained, and the research positioning is not clear.
5. The marking of citations is confusing.
6. In section 2.1, the division of two types of areas lacks quantitative indicators such as population density, building density, or public transport demand.
7. In section 4.1, how are the two study areas A and B selected? Why did you choose these two regions?
8. The evaluation indicators in Table 1 did not consider economic costs and benefits, such as ticket price, fuel cost, labor cost, etc.
9. Section 5.3, the discussion lacks comparative analysis with existing research results.
10. If the author can draw a conclusion on the quantitative relationship between the indicators, such as population, bus demand, operation plan, and benefit, the study will be more meaningful and practical.
Author Response

(The authors gave the same response as above.)
